# DIVINE BENEVOLENCE IS AN $x^2$: GLUS SCALE ASYMPTOTICALLY FASTER THAN MLPS

**Alejandro Francisco Queiruga**
`alejandro.queiruga@gmail.com`

## ABSTRACT

Scaling laws can be understood from ground-up numerical analysis, where traditional function approximation theory can explain shifts in model architecture choices. GLU variants now dominate frontier LLMs and similar outer-product architectures are prevalent in ranking models. The success of these architectures has mostly been left as an empirical discovery. We apply the tools of numerical analysis to expose a key factor: these models have an $x^2$ which enables *asymptotically* faster scaling than MLPs. GLUs have piecewise quadratic functional forms that are sufficient to exhibit quadratic order of approximation. The $L(P)$ scaling slope is $L(P) \propto P^{-3}$ for GLUs but only $L(P) \propto P^{-2}$ for MLPs. We provide a parameter construction and empirical verification of these slopes for low dimension function approximation on synthetic and real data. From the first principles we discover, we make one stride and propose the "Gated Quadratic Unit" which has an even steeper $L(P)$ slope than the GLU and MLP. This opens the possibility of architecture design from first principles numerical theory to unlock superior scaling in large models. Replication code is available at `https://github.com/afqueiruga/divine_scaling`.

## 1 INTRODUCTION

In contemporary LLMs, variants of GLUs (Dauphin et al., 2017) are now the norm (e.g., Gemma (Team et al., 2025) and Qwen (Yang et al., 2025)) and SOTA recommendation and ranking models similarly incorporate outer-product architectures (e.g., Wukong (Zhang et al., 2024) and Deep & Cross Networks (Wang et al., 2017)). Famously, the success of GLUs was an empirical observation attributed to "divine benevolence" (Shazeer, 2020). This work proposes a new understanding through a numerical function-approximation lens for the GLU's empirical success: GLUs form piecewise quadratic functions, over MLPs' piecewise linear representation.

Scaling laws connect to the concept of convergence in numerical analysis. In scientific software, the expectation of exact log-log slopes from derivation is even measured implementation validation (a "convergence test"). When looking at a numerical method, an intuition is to look at the polynomial order of the method's underlying function approximation. Balestriero & Baraniuk (2020) used this insight to propose the Max Affine Spline Operator (MASO) interpretation of ReLU MLPs as piecewise linear splines. We build upon this interpretation to understand the GLU:

$$GLU(x) = d + \sum_{i=0}^{n} D_i \text{relu}(G_i x + g_i) * (U_i x + u_i) \tag{1}$$

and observe when the activation for the $i$-th neuron is "open", its contribution can be expanded as

$$\text{Active Neuron}_i(x) = D_i(G_i U_i x^2 + (g_i U_i + G_i u_i)x + g_i u_i). \tag{2}$$

This makes it apparent that the GLU is a collection of quadratic basis functions with coefficients $D_i$. (We focus on ReLU activations, but GeGLUs and SwiGLUs also exhibit $\sigma(x) \to x$ upon activation.) Plotting randomly initialized networks in Fig. 1 makes this visually apparent. We derive a scaling-optimal construction of network parameters in 1D using this basis function analysis lens, then empirically corroborate the expected scaling slopes for GLUs versus MLPs. We find that for an MLP the error scales like $1/n^2$ (doubling parameters quarters the error), whereas for a GLU it scales like $1/n^3$ (doubling parameters reduces error by a factor of eight). The main text provides theory and empirical results in 1D. Extension to other empirical settings is provided in the appendix.

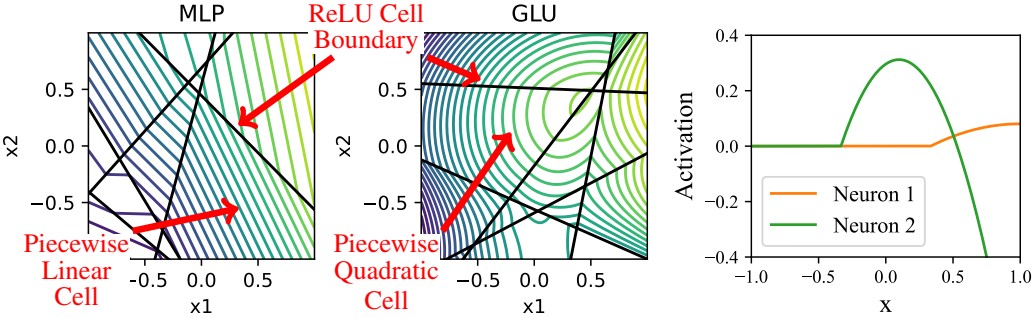

Figure 1: Illustration of the function space of randomly initialized MLPs and GLUs with ReLU activations. Left and middle: randomly initialized MLP and GLU on $\mathbb{R}^2 \to \mathbb{R}$. Black boundaries are the activation boundaries that break the domain into piecewise linear (MLP) or piecewise quadratic partitions. Right: in 1D, each neuron of the GLU forms a single piecewise quadratic function.

## 2 BACKGROUND

There has been extensive literature on understanding the accuracy of neural networks. The empirical scaling law form made famous in Hoffmann et al. (2022) is:

$$L(D, P) = E + \frac{A}{P^\alpha} + \frac{B}{D^\beta} \tag{3}$$

with parameter count $P$, training data size $D$, baseline error $E$, and curve fit parameters $A$, $\alpha$, $B$, and $\beta$. (We will derive an $A$ and $\alpha$.) Other forms have been proposed, e.g. (Li et al., 2025). Scaling laws have been demonstrated in multiple domains, e.g. autonomous driving (Baniodeh et al., 2025). While scaling behavior and architecture design is mostly an empirical art, there is extensive literature in approximation errors and scaling of NNs. Infinite limits of NNs have been analyzed theoretically (Neal, 1996; Lee et al., 2017). Derivations of approximation rates can be found in a few works in the literature: Barron (1994) proved a closed form error bound for sigmoid NNs as a function of width, $e = O(C_f^2/n) + O((nd_{dim} \log N_{data})/N_{data})$ and estimates $L(P) = 1/P$ for sigmoid networks. Telgarsky (2015) demonstrated constructions of deep ReLU networks were exponentially more expressive than shallow networks for a pathological discontinuous classification problem. Hanin & Sellke (2017) provided error bounds for function reconstructions for deep relu networks. The theoretical relu constructions of Yarotsky (2017) included a lower bound of $L \propto 1/n^2$ for single layer relu networks. Bahri et al. (2024) analytically determined four different regimes of scaling laws in large data, large parameter, and underparameterized regimes. They develop a $L(P) = \mathcal{O}(n^{-1})$ for linearized infinite width neural networks. Recommendation systems motivated the outer product forms $y_k = W_{ijk} x_i x_j + b_k$ as "mixing" different features, e.g. explicitly crossing a document embedding against a user profile embedding (Wang et al., 2017; Zhang et al., 2024). We show that even for *scalar* problems the quadratic structure gives faster scaling.

## 3 THEORETICAL DERIVATION OF SCALING SLOPES

We derive an expectation for the MLP and the GLU on 1D function approximation for $L(P)$ in the limit of excessive data and training resources, $D \to \infty$. The number of neurons is related to the parameter count by $P_{MLP} = (dim_x + dim_y + 1)n + dim_y$ and $P_{GLU} = (2dim_x + dim_y + 2)n + dim_y$. Let $f(x)$ denote the target function and $y(x)$ denote the NN approximation. Consider a 1D dataset of points $\{x, f(x)\}$ with $x \in [0, 1]$ where the goal is to reconstruct the function $f$ using a model $y$ using the Root Mean Squared Error (RMSE). If the datapoints $x$ are sampled uniformly in the domain, in the infinite data limit, the error approaches an integral:

$$L = RMSE = \left( \frac{1}{|D|} \sum_D (f(x) - y(x))^2 \right)^{1/2} \xrightarrow{|D| \to \infty} \left( \int_0^1 (f(x) - y(x))^2 \right)^{1/2} \tag{4}$$

There is no noise or uncertainty such that the baseline error of the problem is $E = 0$. We provide the prior stated result by constructing a solution of parameters using the spline interpretation.

**Spline Partitioning using ReLU gates**    Firstly, we set the output bias to $d = f(0)$ arbitrarily. We then utilize the ReLUs to construct equally sized partitions of size $h = 1/n$ as follows: Set all gate boundaries to be $G_i = 1$ and the gate biases to be $g_i = -linspace(0, 1, n)$. This forms regularly spaced activations (see Fig. 3) that break the domain into $n - 1$ closed partitions. See the appendix for a full rollout; for the GLU, one partition has the form

$$y_k(x) := d + \sum_{i=0}^{i<k} D_i(x - ih)(U_i x + u_i) \quad for \quad (k-1)h < x \le kh \tag{5}$$

By construction, exactly one new neuron activates as we cross cell boundaries left-to-right. This allows for solving parameters of individual neurons cell-by-cell using the following procedures.

**MLP Parameter Construction**    We solve for the weights $D_i$ such that at each segment, $y(ih) = f(ih)$ for all points $i = 0, 1, ...N$. We do this moving from left to right. Within the leftmost segment, only one neuron is active, so we solve for the only unknown neuron $D_0 h + d = f(h)$, yielding $D_0 = (f(h) - f(0))/h$. At the next partition cell, $D_1 = (f(2h) - f(0) - 2D_0 h)/h$. The construction can be completed by solving cell-by-cell $y_i(ih) = f(ih)$ by a single pass recurrence relation

$$D_i = (f(ih) - f(0))/h - y_{i-1}(ih). \tag{6}$$

We continue this procedure to the last spline segment to construct an MLP that linearly interpolates at equally spaced node boundaries $x = ih$. The local truncation error $\tau_i(x) = y_i(x) - f(x)$ is

$$\tau_0(x) = \frac{x^2 f_0''}{2} - \frac{hx f_0''}{2} + \frac{x^3 f_0'''}{6} - \frac{h^3 x f_0''''}{24} + \mathcal{O}(h^4 f_0''') \tag{7}$$

The total error $L$ can be estimated by integrating the truncation error of a representative cell $\bar{\tau}$,

$$L = \left( \int_0^1 \tau(x)^2 dx \right)^{1/2} = \left( \sum_{i=0}^n \int_{(i-1)h}^{ih} \tau_i(x)^2 dx \right)^{1/2} \propto \left( \frac{1}{h} \int_0^h \bar{\tau}^2(x) dx \right)^{1/2} \tag{8}$$

Performing this integral using $\tau_0$ as the representative cell and keeping only the leading term yields

$$L_{MLP} \propto \frac{h^2 C_2}{\sqrt{120}} + \mathcal{O}(h^3) \propto \frac{1}{P^2} \tag{9}$$

where $C_2$ is a bound on $f''(x)$ arising from the $f_0''$ in $\bar{\tau}$. As $h = 1/n \propto 1/P$, we obtain $L(P) \propto 1/P^2$, and demonstrate MLP can have the same approximation order as a linear spline.

**GLU Parameter Construction**    The additional free parameters $U_k$ and $u_k$ within cell $k$ can eliminate an additional term in the local truncation error. The recurrent formula for the GLU splines in cell $i$ is $y_i(x) = y_{i-1}(x) + (D_i U_i) x^2 + (D_i U_i g_i + D_i u_i) x + (D_i g_i u_i)$. When lining up $y_i(ih) = f(ih)$, we solve for $u_i$ instead of $D_i$. For the first cell to solve $u_0$,

$$f(h) = D_0 h(U_0 h + u_0) + d \rightarrow u_0 = \frac{-D_0 U_0 h^2 - f(0) + f(h)}{D_0 h} \tag{10}$$

which leaves $D_0$ and $U_0$ free. These are set by minimizing the truncation error within the cell, $\tau_0(x) = -D_0 x(U_0 x + u_0) - f(0) + f(x)$. Substituting $u_0$ and expanding $f(x)$ and $f(h)$ yields

$$\tau_0(x) = D_0 U_0 hx - D_0 U_0 x^2 - \frac{h^2 x f_0'''}{6} - \frac{hx f_0''}{2} + \frac{x^3 f_0'''}{6} + \frac{x^2 f_0''}{2} + \mathcal{O}(h^4 f_0'''') \tag{11}$$

With the two degrees of freedom remaining, we are able to cancel out two terms here by setting

$$U_0 = f_0''/2D_0 \tag{12}$$

with one extra degree of freedom. The truncation error then becomes

$$\tau_0(x) = -\frac{h^2 x f_0'''}{6} + \frac{x^3 f_0'''}{6} + \mathcal{O}(h^4 f_0'''') \tag{13}$$

This forms another recurrence procedure for constructing the GLU parameters: sweep cell by cell by solving $y_i(ih) = y_{i-1}(ih)$ for $u_i$, and then solve for $U_i D_i$ that cancels out the $f_0''$ terms in the truncation error. This systematically bounds the local truncation error by $\tau(x) = \mathcal{O}(h^3 f_0''')$. The global RMSE, again obtained as the square root of the square integral, yields

$$L_{GLU}(P) \propto \frac{h^3 C_3}{\sqrt{945/2}} + \mathcal{O}(h^4) \propto \frac{1}{P^3} \tag{14}$$

where $C_3$ is a bound on $f'''(x)$. Thus, we demonstrate that we can construct a parameterization for the GLU that has the same error rate as a piecewise quadratic spline, with a scaling law $L(P) \propto 1/P^3$.

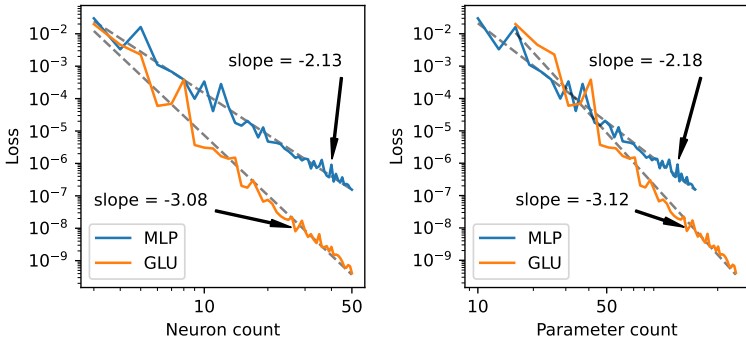

Figure 2: Experimental results for $L(P)$ scaling on 1d function approximation. The x-axis on the left uses neuron count which corresponds to the number of spline knots; changing the variable to parameter count on the right translates the log-log lines but does not change the log-log slope.

## 4    EMPIRICAL MEASUREMENT OF SCALING SLOPES

We fit the target function $f(x) = (1 + \cos^2(\pi x))^{-1}$ on 10k points sampled on $[-1, 1]$. (The higher order Taylor terms of this function decay slowly.) We utilize the spline-based initialization scheme, $G_i = \pm 1$, and $g_i = \pm linspace(-1, 1, n)$, with alternating signs to avoid the dead neurons (see Fig. 3 in the appendix.) The other parameters are initialized with a normal distribution of $U_i, u_i, D_i, d_i \sim \mathcal{N}(0, 1)$. To reach the lowest possible error to extend the convergence plot as far as possible, the models are trained in double precision on CPU using Newton's method, employing layer-by-layer full batch updates, Jacobi preconditioning, singular row elimination, and line search.

Fig. 2 displays the results of the scaling estimation. The NNs optimize on an MSE loss and report the Root Mean Square Error (RMSE) for each width starting from $n = 1$ up to $n = 50$. Using RMSE instead of MSE matches local truncation order and numerical analysis theory. Given all of these data points, we perform a log-log fit. The actual errors are lower than the analytical construction, but bounded by the same rate. The measured slopes match the analytical expectations: 1) The MLP measures a slope of $n^{-2.13}$ and $P^{-2.18}$, matching the expected rate of -2. 2) The GLU measures a slope of $n^{-3.08}$ and $P^{-3.12}$, matching the expected rate of -3. The appendix contains experiments for higher dimensional synthetic and real problems, and an a priori designed Gated Quadratic Unit that achieves a steeper scaling slope.

## 5    CONCLUSION

We systematically explained how to effect different $L(P)$ scaling slopes in NNs, which to our knowledge has not been published previously. Using tools from classical numerical analysis, we show that higher-order approximation behavior appears in modern architectures, offering a lens for architecture design. We posit that the superior order of approximation we demonstrated in GLUs may be a factor that caused them to win out against MLPs. From first principles, we designed the GQU that has an even faster $L(P)$ scaling rate than the GLU and MLP. We hypothesize that scaling large architectures may resemble mixed-method scientific simulations: doubling GLU width could require scaling other components at a different rate to maintain overall efficiency. Our measurements are limited to slopes on 1D synthetic problems with shallow models. This effect may not dominate in large models on real-world datasets. Higher-order scaling requires sufficiently smooth targets; real circuits may not be smooth enough to benefit. The extra nonlinearities can also change optimization dynamics, preventing convergence to spline-optimal minima. This perspective still highlights *efficient* approximation away from infinite width: GLUs can represent curved decision boundaries and complex circuits to the same error with 15 neurons versus 50. Follow-up work should validate convergence rates on real-world datasets and generalize the construction to arbitrary dimensions.

ACKNOWLEDGMENTS

The author thanks David Hansul Park for discussing the initial idea and providing feedback on experiment design.

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

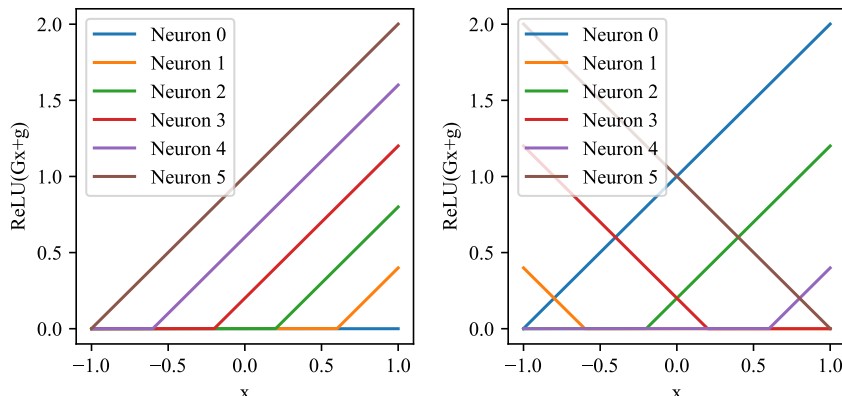

Figure 3: ReLU activations for the uniform cell spline initializations $G = \pm 1$ and $g = \pm linspace(-1, 1, n)$. On the left is an initialization where all face the same way, which simplified the analytical construction. On the right is an alternating construction with the same partitioning but prevents all dead neurons.

Dmitry Yarotsky. Error bounds for approximations with deep relu networks. *Neural networks*, 94: 103–114, 2017.

Buyun Zhang, Liang Luo, Yuxin Chen, Jade Nie, Xi Liu, Shen Li, Yanli Zhao, Yuchen Hao, Yantao Yao, Ellie Dingqiao Wen, et al. Wukong: Towards a scaling law for large-scale recommendation. In *International Conference on Machine Learning*, pp. 59421–59434. PMLR, 2024.

## A  PIECEWISE SPLINE EXPANSIONS

Each cell references an overlapping set of parameters, as the neural network function approximation is not constructed as compact cells (as are splines or partition of unity methods), but the combination of all active neurons in a cell can be analyzed as a piecewise spline, albeit with an awkward parameterization. The piecewise spline construction of the MLP that can be verbosely written out as a conditional, where each row corresponds to a different cell.

$$
y(x) = \begin{cases}
y_{oob} := d & ; & x \le 0 \\
y_0 := d + D_0 x & ; & 0 < x \le h \\
y_1 := d + D_0 x + D_1(x - h) & ; & h < x \le 2h \\
\dots \\
y_k := d + \sum_{i=0}^{i \le k} D_i(x - ih) & ; & kh < x \le (k+1)h \\
\dots \\
y_n := d + \sum_{i=0}^{i < n} D_i(x - ih) & ; & nh < x
\end{cases}
\tag{15}
$$

$y_{oob}$ denotes when $x$ is "out of bounds" of all of the ReLUs and the function is piecewise constant. The analytical construction in the main text solves these cells row by row for $D_i$.

Putting all gates facing in the same direction simplifies the presentation of the construction. In practice the signs on some gates can be flipped to fully cover $\mathbb{R}$, which prevents the leftmost edge of the domain from becoming dead. This construction and one equivalent alternative are plotted in Fig. 3.

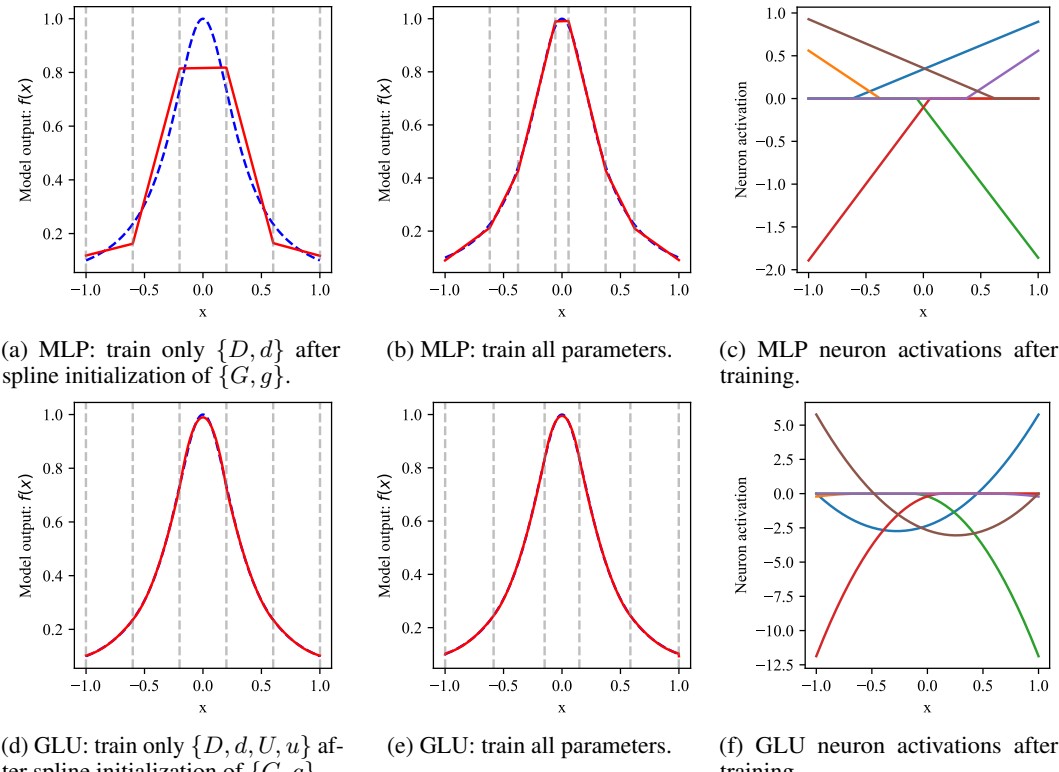

(a) MLP: train only $\{D, d\}$ after spline initialization of $\{G, g\}$.

(b) MLP: train all parameters.

(c) MLP neuron activations after training.

(d) GLU: train only $\{D, d, U, u\}$ after spline initialization of $\{G, g\}$.

(e) GLU: train all parameters.

(f) GLU neuron activations after training.

Figure 4: Comparison of MLP and GLU fitting behavior under spline-based initialization (a,d) and full training (b, e). Vertical lines denote cell boundaries formed by $Gx + g = 0$. Partial optimization in (a,d) already achieves the same asymptotic scaling rate as the analytical construction. The activations of individual neurons of the fully trained MLP and GLU are shown in (c,f).

For the GLU, the partitions are constructed the same, but the spline is piecewise quadratic:

$$
y(x) = \begin{cases}
y_{oob} := d & ; & x \leq 0 \\
y_0 := d + D_0 x (U_0 x + u_0) & ; & 0 < x \leq h \\
y_1 := d + D_0 x (U_0 x + u_0) + D_1 (x - h)(U_1 x + u_1) & ; & h < x \leq 2h \\
\dots \\
y_k := d + \sum_{i=0}^{i \leq k} D_i (x - ih)(U_i x + u_i) & ; & kh < x \leq (k+1)h \\
\dots \\
y := d + \sum_{i=0}^{i < n} D_i (x - ih)(U_i x + u_i) & ; & nh < x
\end{cases}
\tag{16}
$$

The analytical construction in the main text also solves these equations row by row.

## B  EMPIRICAL SPLINE BEHAVIOR

The optimization process is illustrated in Fig. 4 for a 6 neuron MLP and GLU to approximate the target function (blue line). The analytical construction keeps $G$ frozen and evenly spaces the cells formed by the activation functions, equivalent to the leftmost panels in both rows. In the construction, the function value at knot boundaries (vertical dashed lines) lie on the target function; with MSE optimization they do not. The analytical solutions and those in (a,d) achieve the theoretical optimal scaling slopes, even without achieving the lowest possible MSE. Fully training the network yields the approximation in the middle panels and the neuron activations in the right panels.

As a baseline, we compare the NNs to linear and quadratic splines, also implemented in the same PyTorch training script, in Fig. 5. We see that the error of the MLP and GLU is comparable to the spline counterparts. The MLP and GLU achieve slightly smaller errors than their spline counterparts.

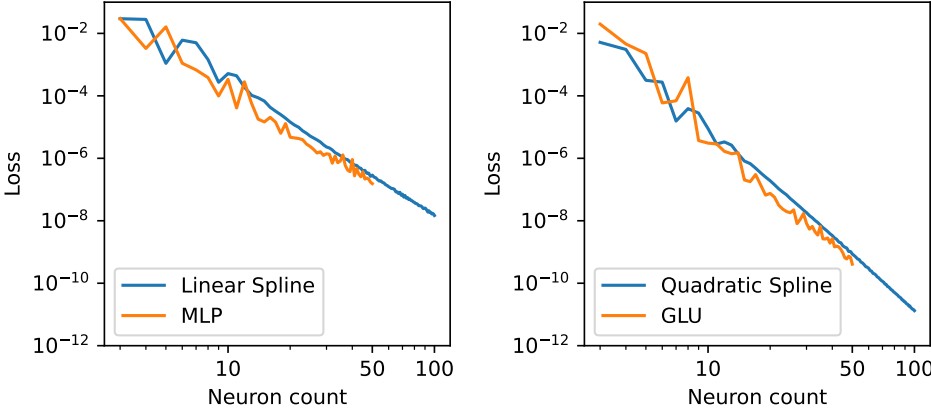

Figure 5: MLPs and GLUs have similar approximation errors and order of convergence as their spline counterparts. Neuron count is analogous to the number of knots (control nodes) in a spline.

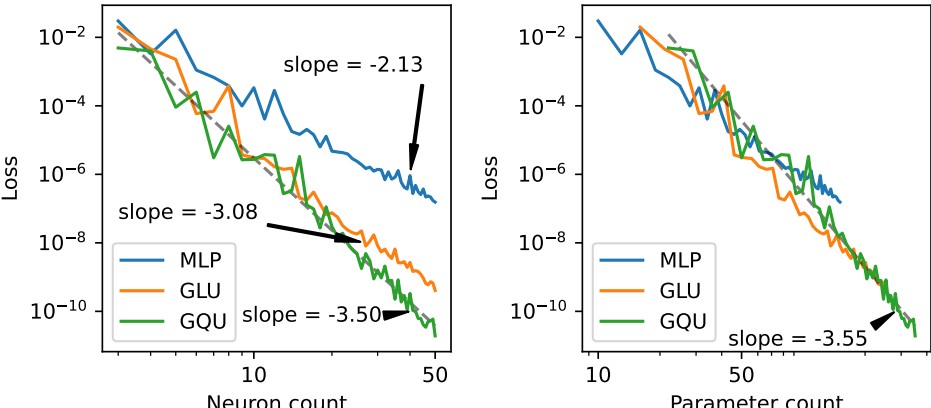

Figure 6: Novel architecture design to scale faster than a GLU: the Gated Quadratic Unit (GQU) has faster scaling than a GLU at $L(P) \propto P^{-3.5}$.

The additional degrees of freedom can move spline knots enabling a modest *constant factor* reduction in error.

## C   FIRST PRINCIPLES ARCHITECTURE DESIGN FOR SCALING: GATED QUADRATIC UNIT

Given the confirmation that numerical analysis techniques apply to ML architectures, it is possible to construct a priori design architectures with faster scaling properties. From first principles, we propose the Gated Quadratic Unit,

$$GQU(x) = d + D(act(Gx + g) * (Ux + u) * (Qx + q)) \tag{17}$$

which has cubic terms in its unfolding. Ideally, we expect it to be possible to have $L(P) \propto P^{-4}$. Repeating the empirical analysis from the main text, we observe a slope of -3.5 in Fig. 6. This is lower than the gut instinct hypothesis, but still faster than the GLU, opening up the possibility for a new principled methodology for deriving new architectures.

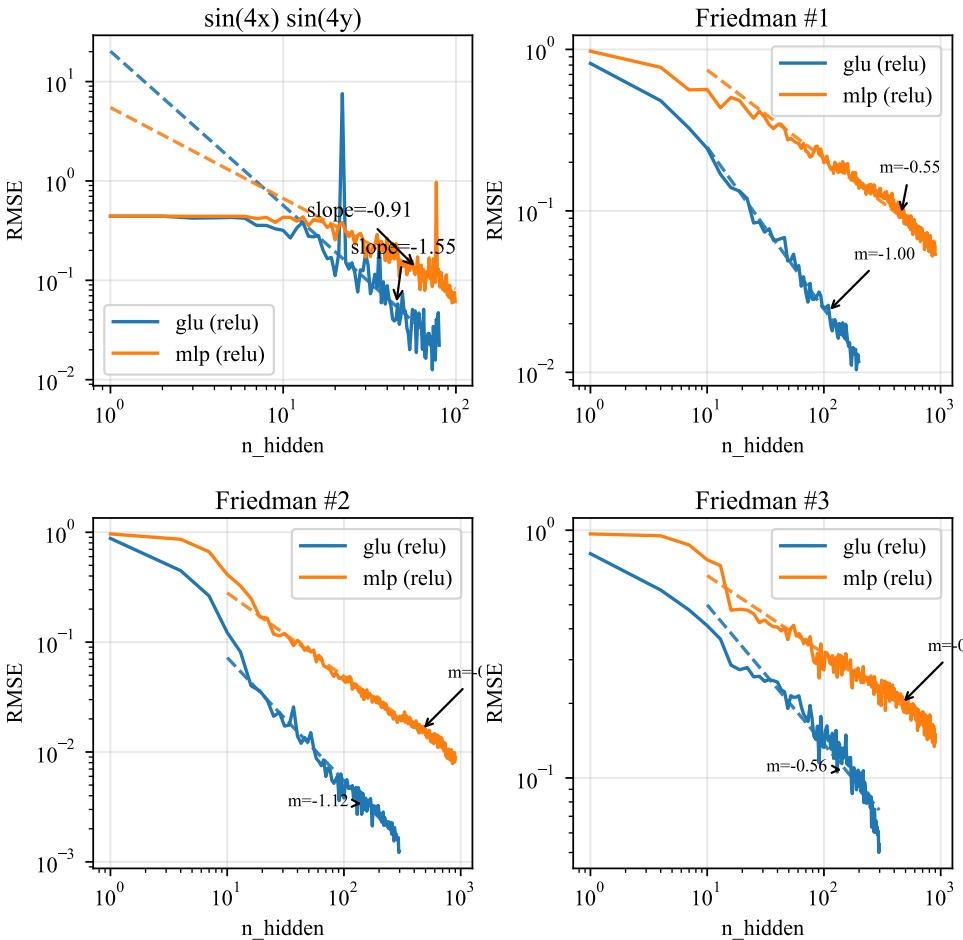

Figure 7: Higher dimension synthetic problems. See the measured log-log slopes in Table 1.

## D HIGHER DIMENSIONAL SYNTHETIC AND REAL PROBLEMS

We empirically verify the different scaling slopes in higher dimensions for synthetic problems and real world regression datasets. We use the Friedman synthetic problems from sklearn and two datasets from the UCI dataset, California housing and Airfoil Self-drag. The results are plotted in figures 7 and 8 and the measured slopes are summarized in table 1. We observe that the scaling slopes for the GLU are steeper than the MLP for synthetic problems and real world datasets, except for the California housing dataset. As expected, the scaling slope decreases with dimension by a factor of $1/dim_x$. However, not every problem matches the expected slope.

Table 1: Empirically measured slopes.

| Problem | $x$ dim | MLP | GLU | GQU |
|---|---|---|---|---|
| $1/(1 + cos(\pi x)^2)$ | 1 | $-2.13$ | $-3.08$ | -3.55 |
| $sin(4x)sin(4y)$ | 2 | $-0.91$ | $-1.55$ | - |
| Friedman1 | 5 | $-0.55$ | $-1.00$ | - |
| Friedman2 | 4 | $-0.75$ | $-1.12$ | - |
| Friedman3 | 4 | $-0.31$ | $-0.56$ | - |
| Airfoil Self-drag | 5 | $-0.25$ | $-0.39$ | - |
| California Housing | 8 | $-0.09$ | $-0.09$ | - |

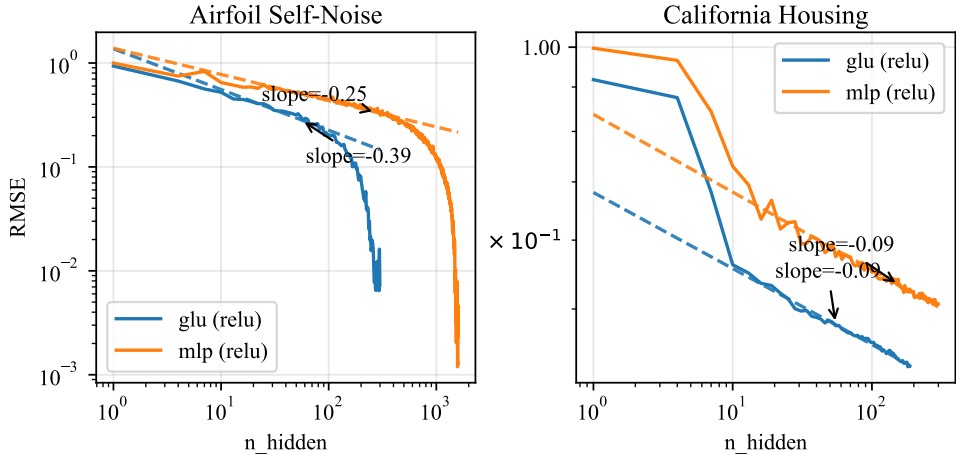

Figure 8: Real world benchmark regression datasets. See the measured log-log slopes in Table 1.

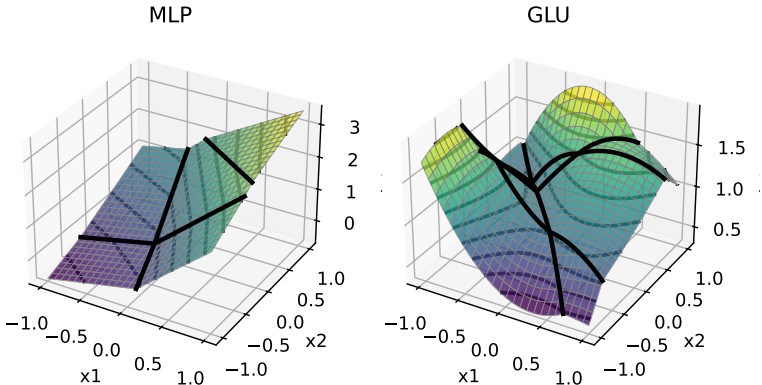

Figure 9: Surfaces of randomly initialized MLP and GLU with four neurons. The black lines demarcate the hinge of the ReLU activations, breaking the surface into piecewise linear and piecewise quadratic regions.

## E   NEURON VISUALIZATIONS OF 2D FUNCTIONS

The partitioning into piecewise splines also applies to higher dimensions, where the activation functions form irregular cells along their hinges. For any set of parameters, the boundary surfaces can be determined by solving for $G_{ij}x_j + g_i = 0$ for each neuron $i$. Randomly initialized $\mathbb{R}^2 \to \mathbb{R}$ networks are shown in Fig. 1 with four neurons. Each neuron's contributions are shown in Fig. 10 and Fig. 11, where the linear versus quadratic nature is apparent.

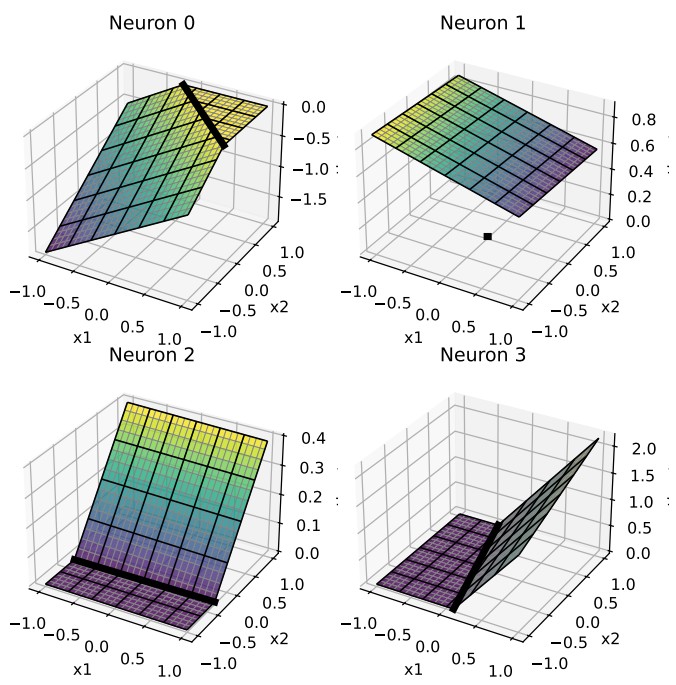

Figure 10: Neuron activations of a randomly initialized MLP with four neurons form a piecewise linear basis set.

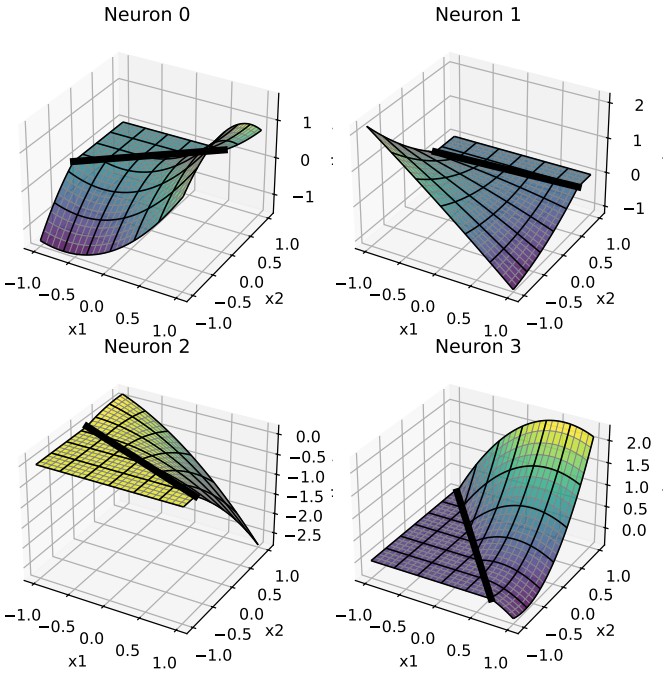

Figure 11: Neuron activations of a randomly initialized GLU with four neurons form a piecewise quadratic basis set.

