# OpenReview forum: "Divine Benevolence is an $x^2$:  GLUs have asymptotically faster scaling laws than MLPs"
_ICLR.cc/2026/Workshop/Sci4DL — Sci4DL 2026_

### Official Review · Reviewer_m5hy · 2026-02-25

**Fit:** 3
**Significance:** 1
**Confidence:** 2

**Summary:**

The authors explore a numerical approach to understanding scalling laws particularly observing how they change through the use of different inductive biases provided by MLP’s and GLU’s. They present evidence that the scaling of a GLU is $x^{2}$ over a typical MLP  on a 1-D synthetic task.

**Strengths:**

1. The setup of the synthetic experiment is insightful and helps to support the theoretical work.
2. Providing a numerical basis for the scaling laws of activation functions is an interesting research direction which has the potential to provide widespread insights.
3. The parameter scaling empirical experiments are insightful, however it would be interesting to see neurone count scaled to 100 neuron so that all plotted lines are to the same scale.
4. Theoretical results are appreciated but the empirical results are lacklustre which unfortunately undermine the significance of the paper.

**Suggestions:**

1. I would suggest that the authors tamper some of the claims made in the paper regarding the scaling of the results to LLM's given that the experimental results are only shown on very small architectures on 1-D synthetic tasks. Perhaps bringing forward the caveats included in the conclusion:  **L205** “Our measurements are limited to slopes on 1D synthetic problems with shallow models and” **L206** “This effect may not dominate in large models on real-world datasets” as it would allow the reader to focus more on the results as they are presented instead of considering the limited experimental setup.

2. For the empirical results the optimisation deviates from traditional DNN’s training (Newton’s methods 2nd order derivatives, with layer-by-layer full batch updates, Jacobi preconditioning, singular row elimination and spline-based initialization) which reduces confidence that these findings would scale to other settings without having evidence in traditional optimisation settings. As it is possible to conduct small scale experiments with standard training setups this is a major drawback of the paper.

3. With small scale experiments provided on real world datasets the results would be compelling, however, given the current set up the discussion at the workshop would be limited for this work.

4. The paper states that the code to reproduce the results would be provided on **L186**, however it is not provided.

---

### Official Review · Reviewer_t4my · 2026-02-26

**Fit:** 3
**Significance:** 2
**Confidence:** 2

**Summary:**

This paper derives scaling laws for multi-layer perceptrons with ReLUs and GLUs in the asymptotic limit of unbounded training data, using tools from numerical analysis. Specifically, using the framework of Max Affine Spline Operators (MASO), it shows that the loss scaling slope for GLUs is $P^{-3}$, in contrast to a slope of $P^{-2}$ for ReLUs where $P$ is the number of parameters. The theoretical results are validated on a cosecant function reconstruction problem. This argument is used to justify the empirical effectiveness of GLUs in transformer-based models, which has otherwise been attributed to "divine benevolence".

**Strengths:**

- The paper poses an interesting and relevant question and provides a solid, clean, and interpretable answer to it.
- The illustrations in Figure 1 and the Conclusion are especially helpful in providing intuition and grounding the relevance of the results to practical settings respectively.
- It uses a theoretical tool that is not often seen in the literature, but highlights its potential usefulness to answer this relevant question.

**Suggestions:**

- I suspect some of the equations may have typos. Namely, equation (2) should have a $G_{i}u_{i}$ term instead of the $G_{i}g_{i}$ term. Equation (7) is preceded by a reference to the truncation error $\tau_{i}(x)$ but the equation itself expresses $\tau_{0}(x)$.
-  Some additional background on the MASO interpretation would be much appreciated, but I do see the difficulties of doing so within the main content of the paper, given the page limits.
- It would be quite interesting to understand, even if just empirically, if and how these results change in the realm of limited data i.e. how does the scaling law comparison change when dataset size $D$ is factored in? Perhaps this could be done with the same empirical experiment repeated with different dataset sizes.

---

### Meta-Review · Area_Chair_8Hzj · 2026-03-01

**Recommendation:** Accept

**Metareview:**

Recommending acceptance. Strongly suggest the authors work through and take into account the reviews.

---

### Decision · Program_Chairs · 2026-03-02

Accept